# Composition of early life leukocyte populations in preterm infants with and without late-onset sepsis

Julie Hibbert[1,2,3], Tobias Strunk[1,2,3], Elizabeth Nathan[2], Amy Prosser[2], Dorota Doherty[2], Karen Simmer[2], Peter Richmond[2,3], David Burgner[4,5], Andrew Currie[3,6]*

1 Neonatal Directorate, Child and Adolescent Health Service, Perth, Western Australia, Australia, 2 Medical School, University of Western Australia, Perth, Western Australia, Australia, 3 Wesfarmers Centre of Vaccines and Infectious Diseases, Telethon Kids Institute, Perth, Western Australia, Australia, 4 Murdoch Children's Research Institute, Royal Children's Hospital, Parkville, Victoria, Australia, 5 Department of Paediatrics, University of Melbourne, Melbourne, Victoria, Australia, 6 Centre for Molecular Medicine and Innovation Therapeutics, Murdoch University, Perth, Western Australia, Australia

* A.Currie@murdoch.edu.au

**Data Availability Statement:** All relevant data are within the paper and its Supporting Information files.

## Abstract

### Background

Composition of leukocyte populations in the first month of life remains incompletely characterised, particularly in preterm infants who go on to develop late-onset sepsis (LOS).

### Aim

To characterise and compare leukocyte populations in preterm infants with and without LOS during the first month of life.

### Study design

Single-centre prospective observational cohort study.

### Participants

Infants born <30 weeks gestational age (GA).

### Outcome measures

Peripheral blood samples were collected at 1, 7, 14, 21 and 28 days of life. Leukocyte populations were characterised using 5-fluorophore-6-marker flow cytometry. Absolute leukocyte counts and frequency of total CD45+ leukocytes of each population were adjusted for GA, birth weight z-scores, sex and total leukocyte count.

### Results

Of 119 preterm infants enrolled, 43 (36%) had confirmed or clinical LOS, with a median onset at 13 days (range 6–26). Compared to infants without LOS, the adjusted counts and frequency of neutrophils, basophils and non-cytotoxic T lymphocytes were generally lower

**Funding:** This study was funded by grants from the National Health and Medical Research Council of Australia (#572548, www.nhmrc.gov.au), the Western Australia Telethon Channel 7 Trust, the Western Australia Department of Health and the Wesfarmers Centre of Vaccines and Infectious Diseases. DB is supported by a National Health and Medical Research Council of Australia Investigator Grant (#1175744, www.nhmrc.gov.au) and their research at the Murdoch Children's Research Institute is supported by the Victorian Government's Operational Infrastructure Support Program. TS is supported by a Raine Foundation and Western Australia Department of Health Clinician Research Fellowship (rainefoundation.org. au). JH is supported by a University Postgraduate Award and a Wesfarmers Centre of Vaccines and Infectious Diseases Scholarship (www. infectiousdiseases.telethonkids.org.au). There was no additional external funding received for this study. The funders had no role in the study design, data collection and analysis, decision to publish, or preparation of the manuscript.

**Competing interests:** The authors have declared that no competing interests exist.

and immature granulocytes were higher over the first month of life in infants who developed LOS. Specific time point comparisons identified lower adjusted neutrophil counts on the first day of life in those infants who developed LOS more than a week later, compared to those without LOS, albeit levels were within the normal age-adjusted range. Non-cytotoxic T lymphocyte counts and/or frequencies were lower in infants following LOS on days 21 and 28 when compared to those who did not develop LOS.

## Conclusion

Changes in non-cytotoxic T lymphocytes occurred following LOS suggesting sepsis-induced immune suppression.

## Introduction

Late-onset sepsis (LOS; onset >72 hours of age) remains a frequent cause of morbidity and mortality in preterm infants (born <37 weeks gestation age (GA)) [1]. Risk is inversely related to GA and birth weight and LOS occurs in over 20% infants born ≤32 weeks GA [2]. Late-onset sepsis and the associated inflammation contribute to increased mortality and adverse outcomes, particularly impaired neurodevelopment [3].

Leukocytes play important roles in initiating inflammatory responses, stimulating complement pathways and activating adaptive immune responses. However, composition of early life leukocyte populations remain incompletely characterised, particularly in preterm infants who go on to develop LOS [4, 5]. Most data on neonatal leukocyte numbers and/or frequencies are derived from cord blood studies, from which immune parameters may be poorly reflective of postnatal immunity [6]. Moreover, leukocyte data are routinely obtained from manual blood films and automated blood count analysers and lack information on relative leukocyte frequencies [7, 8]. Consequently, there are few reports on leukocyte numbers and proportions preceding and following neonatal sepsis [4]. In this prospective observational study, we characterised the principal leukocyte populations during the first month of life in very preterm infants (born <30 weeks GA) with and without LOS. We hypothesised that infants who subsequently developed LOS would have differential leukocyte counts and proportions compared to uninfected infants, both preceding and following sepsis.

## Methods

### Study population

This prospective, longitudinal observational study of innate immune system development during the first month of life in very preterm infants born <30 weeks GA was approved by the Women and Newborn Health Services Human Research Ethics Committee at King Edward Memorial Hospital, Perth, Australia (1627/EW) [9–11]. Written, informed consent was obtained from parents prior to their child's participation. Infants born at <30 weeks GA without major congenital anomalies were eligible for recruitment to this study between July 2009 and October 2011.

In accordance with local clinical guidelines, blood culture was performed if infants had clinical signs suggestive of LOS, including increased ventilation or oxygen requirements, lethargy, decreased perfusion, temperature instability, vomiting, feed intolerance, increased apnoea and/or bradycardia. LOS was defined as follows: 'Confirmed LOS'—positive blood culture and

a C-reactive protein (CRP) of >15 mg/L within 72 h of blood culture. 'Clinical LOS'—negative blood culture and a CRP of >15 mg/L within 72 h of blood culture [9, 12]. 'Any LOS'—infants with confirmed and clinical LOS. Infants with a Gram-positive blood culture (coagulase-negative staphylococci n = 2 and *Enterococcus faecalis* n = 1), 2–4 serial CRPs of <15 mg/L within 72 h of blood culture sampling and the absence of sustained clinical features of LOS were considered contaminants and analysed with infants that did not develop confirmed or clinical LOS ('no LOS').

Relevant demographic and clinical data until discharge from the neonatal intensive care unit were extracted from electronic patient databases. Placental histology was examined as part of routine care for infants born <30 weeks GA, using an adaption of a widely accepted semi-quantitative scoring system, as previously described [13].

## Whole blood sampling

Peripheral venous blood samples (0.8 mL) were collected into Lithium-heparin tubes (BD Biosciences, North Ryde, NSW, Australia) on days 1 (n = 119), 7 (n = 111), 14 (n = 109), 21 (n = 111) and 28 (n = 111) of life and processed within 4 h of collection to characterise immune development using a number of assays [9–11].

## Leukocyte immunostaining for flow cytometry

For this study, 50 μL whole blood samples were stained with leukocyte marker antibodies using an established 5-fluorophore-6-marker surface staining method, as previously described [14]. In brief, whole blood was incubated with a cocktail of monoclonal mouse anti-human antibodies: CD2-APC, CD36-PE, CD16-APC-H7, CD45-AmCyan, CD19-V450 and rat anti-human chemoattractant receptor-homologous molecule expressed on T helper type 2 (Th2) cells (CRTh2)-AF647 (all BD Biosciences). Red blood cells were lysed with BD FACS Lysing Solution before fixation in Stabilizing Fixative and transfer to TruCOUNT Tubes (all BD Biosciences).

## Flow cytometry analysis

Flow cytometry analysis was performed as previously described [9]. Briefly, data was acquired in real-time on a triple-laser eight-colour FACSCanto II flow cytometer with FACSDiva software (BD Biosciences). Flow cytometer fluorescence intensities were monitored weekly with SPHERO Rainbow Calibration Beads (SheroTech Inc, Lake Forest, IL, USA). An established gating strategy, validated in neonatal cord blood and critically ill newborns, was used to identify principal leukocyte populations using FlowJo version 8 software (Tree Star Inc., Ashland, OR, USA) [14–16]. Leukocytes were separated from debris using positivity for the common leukocyte antigen CD45. Through successive gating, the following populations were identified: neutrophils (SSC$^{high}$/CD16$^+$), immature granulocytes (SSC$^{high}$/CD16$^-$/CD2$^-$/CRTh2$^-$), eosinophils (SSC$^{high}$/CD16$^-$/CRTh2$^+$), basophils (SSC$^{int}$/CD16$^-$/CRTh2$^+$), CD16$^-$ classical monocytes (SSC$^{int}$/CD16$^-$/CD2$^-$/CRTH2$^-$/CD19$^-$/CD36$^+$), CD16$^+$ non-classical monocytes (SSC$^{int}$/CD16$^+$/CD2$^-$/CRTH2$^-$/CD19$^-$/CD36$^+$), B lymphocytes (SSC$^{low}$/CD16$^-$/CD2$^-$/CRTH2$^-$/CD36$^-$/CD19$^+$), non-cytotoxic (CD16$^-$) T lymphocytes (SSC$^{low}$/CD16$^-$/CD2$^+$/CRTH2$^+$) and cytotoxic (CD16$^+$) T lymphocytes and natural killer (NK) cells (SSC$^{low}$/CD16$^+$/CD2$^+$/CRTH2$^+$), as shown in S1 Fig in S1 File. Based on the surface markers used, we speculate that the neutrophil population consists mainly of mature neutrophils and less-mature banded cells (CD16$^{dim}$), the immature granulocytes includes promyelocytes, myelocytes and metamyelocytes, the CD16$^+$ cytotoxic T/NK lymphocytes contains mostly effector CD8$^+$ T cells, and the CD16$^-$ non-cytotoxic T lymphocytes are mainly CD4$^+$ T cells [17–23]. Other cells in the T cell compartment,

such as γδ T and NKT cells are likely to be included in the non-cytotoxic and cytotoxic T/NK lymphocyte populations [24].

## Statistical analysis

Continuous data were summarised as median and interquartile ranges (IQR, 25th-75th) and categorical data with frequency distributions. Group (any LOS or no LOS) allocation was made based on the whole study period (i.e. infants who developed LOS during the study period were assigned to the LOS group from Day 1). Univariate comparisons of demographic and clinical data between the any LOS and no LOS groups, and confirmed LOS and clinical LOS groups, were performed using the Mann-Whitney test for continuous outcomes and Fisher's exact test for categorical outcomes. Absolute leukocyte counts were calculated according to BD TruCOUNT instructions by dividing the number of events in the population of interest by the total bead count and multiplying the number of beads per test, divided by the sample volume with dilutions accounted for [9]. Descriptive summaries, prior to analysis, of absolute leukocyte populations counts and frequency of total CD45+ leukocytes data are presented in S1 Table in S1 File. Leukocyte count and frequency data were transformed to the natural logarithm for analysis to fulfil the assumption of normality of the residuals. Mixed linear regression analysis using a random effects model and a group by time interaction was performed to assess the effect of LOS on leukocyte population levels, while accounting for the repeated measures at time points. All models were adjusted for known confounders, GA, birth weight z-score, sex and total leukocyte count [2]. For weekly analysis, post-hoc pairwise differences in adjusted counts between each time point were calculated within LOS and no LOS groups. Analogous analyses assessed temporal changes in leukocyte counts and frequencies, and the ratio of neutrophil counts to lymphocyte counts between groups. All post-hoc pairwise comparisons were Bonferroni-corrected to maintain an overall alpha error rate of 0.05. Model parameters were back transformed and presented graphically as geometric means and their 95% confidence intervals (CI). Stata (version 16, StataCorp, College Station, TX, USA) statistical software was used for data analysis. Prism 8 for macOS (GraphPad, La Jolla, CA, USA) was used to generate figures. *P* values <0.05 were considered statistically significant.

## Results

### Clinical characteristics of study cohort

Of the 129 very preterm infants recruited to this study, infants who suffered from early-onset sepsis (n = 4) or necrotising enterocolitis (n = 5) during the study period or had insufficient samples (n = 1) were excluded from analysis. Clinical characteristics of the analysed cohort (n = 119) are shown in Table 1. Consistent with known risk factors for LOS, preterm infants with any LOS (confirmed (n = 28) and clinical (n = 15)) were younger and smaller at birth and required more respiratory support compared to infants without LOS (n = 76; Table 1) [2]. The median age at the onset of LOS was postnatal day 13 (IQR 25th-75th 10–18 days; range 6–26 days). Three infants, all without LOS, died on days 2 and 3 due to pulmonary haemorrhage (n = 2) and extreme prematurity (n = 1). Data collected prior to death were included in the analysis and leukocyte populations were within the range of uninfected infants. Most of the confirmed LOS episodes (n = 23, 82.1%) were caused by Gram-positive bacteria, predominantly Coagulase-negative staphylococci (n = 19, 82.6%), followed by *Enterococcus faecalis* (n = 2), *Staphylococcus aureus* (n = 1) and *Bacillus sphaericus* (n = 1). Gram-negative infections (n = 5, 17.9%) were caused by *Escherichia coli* (n = 3), *Enterobacter cloacae* (n = 1) and *Klebsiella pneumoniae* (n = 1). The clinical characteristics of infants with confirmed and clinical LOS

**Table 1. Basic clinical characteristics of the study cohort.**

|  | Any LOS | No LOS | P value |
|---|---|---|---|
|  | n = 43 | n = 76 |  |
| Gestational age (weeks) | 26.0 (25.3–28.3) | 27.7 (26.6–29.1) | 0.0005 |
| Birth weight (grams) | 790 (660–955) | 1010 (810–1240) | <0.0001 |
| Birth weight z-score | -0.05 (-0.75–0.32) | 0.21 (-0.54–0.64) | 0.105 |
| Males | 22 (51.2) | 43 (56.6) | 0.702 |
| Small for gestational age | 4 (9.3) | 4 (5.2) | 0.455 |
| Multiple birth | 9 (20.9) | 15 (19.7) | >0.999 |
| Caesarean section | 25 (58.1) | 42 (55.3) | 0.848 |
| ROM | 17 (39.5) | 24 (31.6) | 0.425 |
| Antenatal steroids | 40 (93.0) | 74 (97.4) | 0.350 |
| Apgar <7 at 5 min | 12 (27.9) | 11 (14.5) | 0.093 |
| Chorioamnionitis^ | 18/40 (45.0) | 31/62 (50.0) | 0.687 |
| Mechanical ventilation | 41 (95.3) | 69 (90.8) | 0.485 |
| Duration (hours) | 306 (41–889) | 21 (11–101) | <0.0001 |
| CPAP | 43 (100) | 73 (96.1) | 0.552 |
| Duration (hours) | 879 (643–1237) | 761 (174–1130) | 0.025 |
| IVH (grade III/IV) | 4 (9.3) | 4 (5.3) | 0.458 |
| Chronic lung disease | 15 (34.9) | 12 (15.8) | 0.023 |
| Postnatal dexamethasone | 8 (18.6) | 3 (3.9) | 0.005 |
| ROP (stage III/IV)^ | 2/40 (5.0) | 0/72 (0) | 0.126 |
| Length of NICU stay (days) | 101 (69–136) | 63 (49–84) | <0.0001 |
| Mortality to 28 days | 0 (0) | 3 (4.0) | 0.552 |
| Age at LOS onset (days) | 13 (10–18) | - | - |
| Highest CRP (mg/L)* | 43 (25–73) | - | - |

Data are expressed as median (IQR) or n (%), as appropriate.

ROM, rupture of membranes >24 hours prior to delivery; CPAP, continuous positive airway pressure; IVH, intraventricular haemorrhage; ROP, retinopathy of prematurity; CRP, C-reactive protein.

^from available reports.

* Within 72 hour of blood culture.

–data not available or applicable.

were similar except for a higher maximum CRP in the confirmed LOS group (S2 Table in S1 File).

## Leukocyte populations during the first month of life

The adjusted counts and frequencies of neutrophil, basophil, immature granulocytes and non-cytotoxic T cells were generally different between infants with and without LOS during the first month of life (Figs 1 and 2; S3 and S4 Tables in S1 File). Neutrophil counts and frequencies were unchanged between days 1–14 and then decreased thereafter in the no LOS group, but remained lower and unchanged over time in the any LOS group. Basophil counts and frequencies peaked by day 7 and decreased thereafter in the no LOS group, yet levels remained the same after increasing from days 1 to 7 in the any LOS group. Non-cytotoxic T lymphocyte counts and frequencies increased with age during the first month of life, but were lower in the any LOS group compared to the no LOS group. Immature granulocyte counts and frequencies were unchanged on days 1–7 in both the any and no LOS groups, but decreased thereafter in the no LOS group and remained higher in any LOS until day 28. The temporal trends in

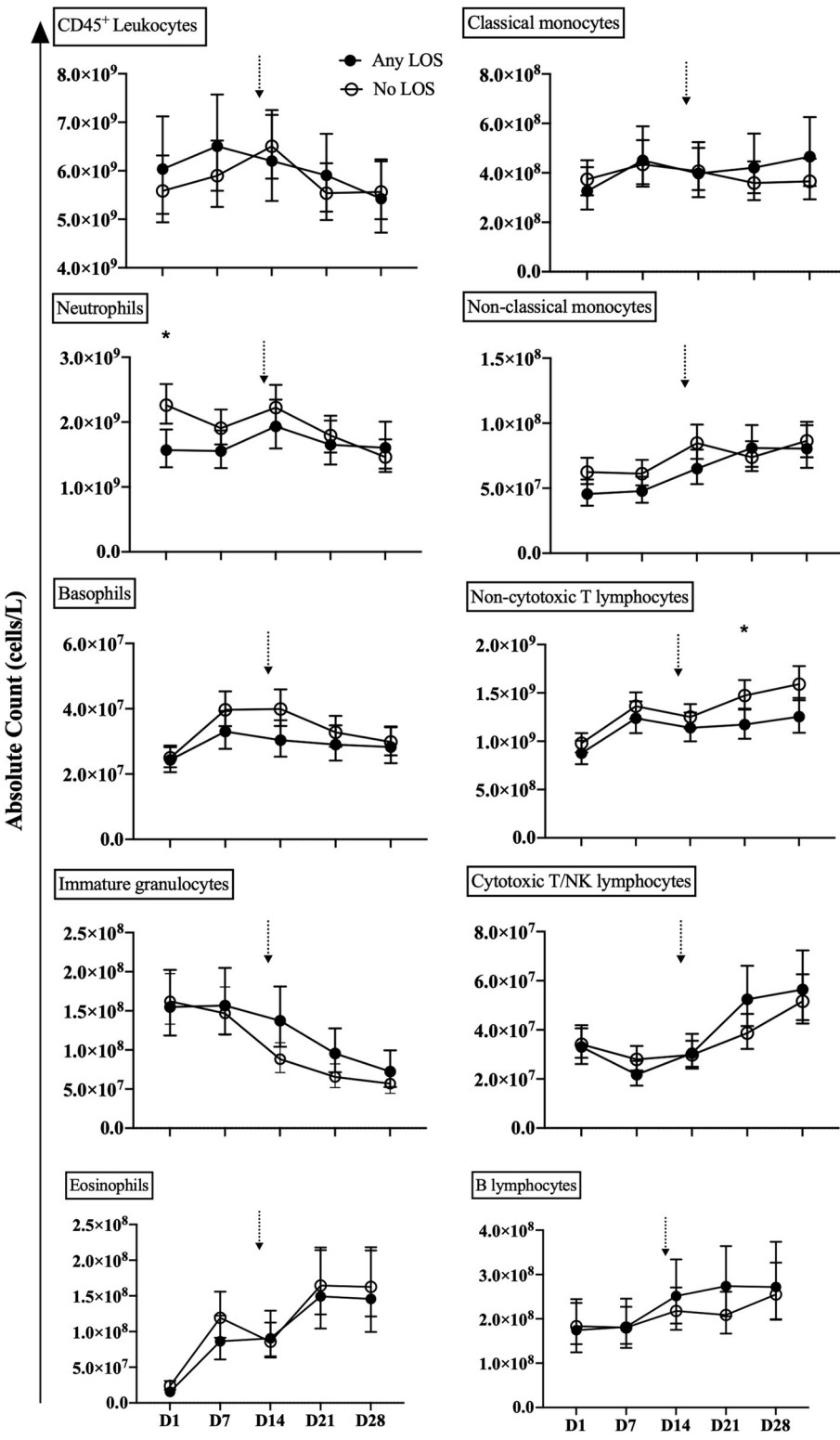

**Fig 1. Absolute leukocyte counts.** The absolute counts of total leukocyte, granulocyte, monocyte and lymphocyte populations from peripheral blood of preterm infants with any LOS (confirmed and clinical; closed circle) and no LOS (open circle). Analysis was performed by mixed linear regression analysis using a random effects model that was adjusted for known confounders, GA, birth weight z-score, sex and total leukocyte count. Data shown as geometric mean with 95% confidence intervals. Arrows indicate the median age of LOS onset of 13 days in infants in the any LOS

group. *Comparison between preterm infants with any LOS and no LOS at the specified time point; level of significance indicated by symbols, e.g. *$P < 0.05$.

counts and frequencies of classical monocytes remained unchanged during the first month of life for both the any and no LOS groups (Figs 1 and 2; S3 and S4 Tables in S1 File). Non-classical monocytes, eosinophils and cytotoxic T/NK and B lymphocytes increased with age and were generally similar between the any and no LOS groups (Figs 1 and 2; S3 and S4 Tables in S1 File).

## Leukocyte populations at each time point

Neutrophil and non-cytotoxic T lymphocyte counts and/or frequencies were associated with LOS. Adjusted neutrophil counts were lower in infants with any LOS than those with no LOS on day 1 (mean $1.57x10^9$/L vs $2.26x10^9$/L, p = 0.011), but were similar at all other time points (Figs 1 and 2). Non-cytotoxic T lymphocyte counts and/or frequencies were lower in the any LOS group compared to the no LOS group on days 21 (mean $1.17x10^9$/L vs $1.47x10^9$/L, p = 0.048; 22.3% vs 27.7%, p = 0.021) and 28 (mean 23.9% vs 30.3%, p = 0.010; Figs 1 and 2). In contrast, counts and frequencies of immature granulocytes, basophils, eosinophils, classical and non-classical monocytes and cytotoxic T/NK and B lymphocytes were similar between the no LOS and any LOS groups at all time points (Figs 1 and 2).

## Neutrophil-to-lymphocyte count ratio

There were no differences in the ratios of neutrophils-to-all-lymphocytes and to lymphocyte sub-populations between the any LOS and no LOS groups (Fig 3).

## Confirmed and clinical LOS

Leukocyte populations and neutrophil-to-lymphocyte ratios were similar between infants with confirmed and clinical LOS except for an elevated frequency in non-cytotoxic T lymphocytes on day 7 in the confirmed LOS group (mean 24.6% vs 17.8%, p = 0.031).

## Discussion

To our knowledge, this is the first study to characterise preterm infant peripheral blood principal leukocyte populations from a single sample taken weekly during the first month of life. We show that neutrophils, basophils and non-cytotoxic T lymphocytes were generally lower and immature granulocytes were higher over the first month of life in infants who developed LOS, compared to infants without LOS. Specific time point comparisons revealed that infants with LOS had significantly lower neutrophils on the first day of life preceding LOS and non-cytotoxic T lymphocytes were decreased following LOS.

### Leukocyte populations during the first month of life

We extend the findings of previous studies that examined the development of monocyte and basophil populations to one week of age by characterising these populations through the first month of life, a time when the risk of sepsis is the highest [25–28]. The counts/frequencies of the granulocyte and monocyte populations observed in our study are consistent with available previous data, obtained mainly from manual blood films and automated analyses, and some flow cytometry-based methods [8, 25, 28–32]. We found counts/frequencies of non-cytotoxic T lymphocytes remained relatively stable during the first month of life, whereas cytotoxic T/

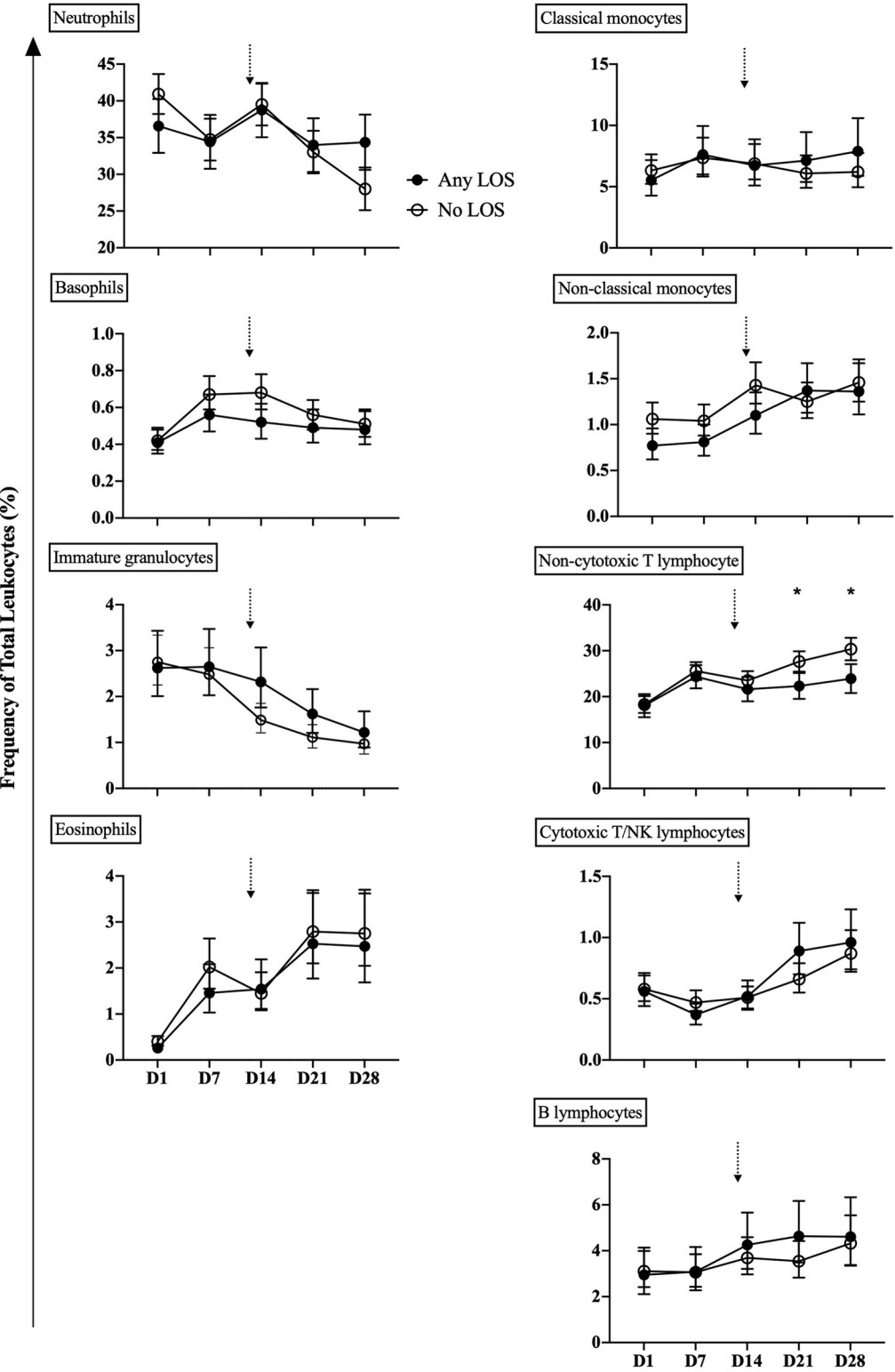

**Fig 2. Leukocyte frequency.** The percentage of total CD45[+] granulocyte, monocyte and lymphocyte populations from preterm infants with any LOS (confirmed and clinical; closed circle) and no LOS (open circle). Analysis was performed by mixed linear regression analysis using a random effects model that was adjusted for known confounders, GA, birth weight z-score, sex and total leukocyte count. Data shown as geometric mean with 95% confidence intervals. Arrows indicated the median age of LOS onset of 13 days in infants in the any LOS group. *Comparison between preterm infants with any LOS and no LOS at the specified time point; level of significance indicated by symbols, e.g. *$P < 0.05$.

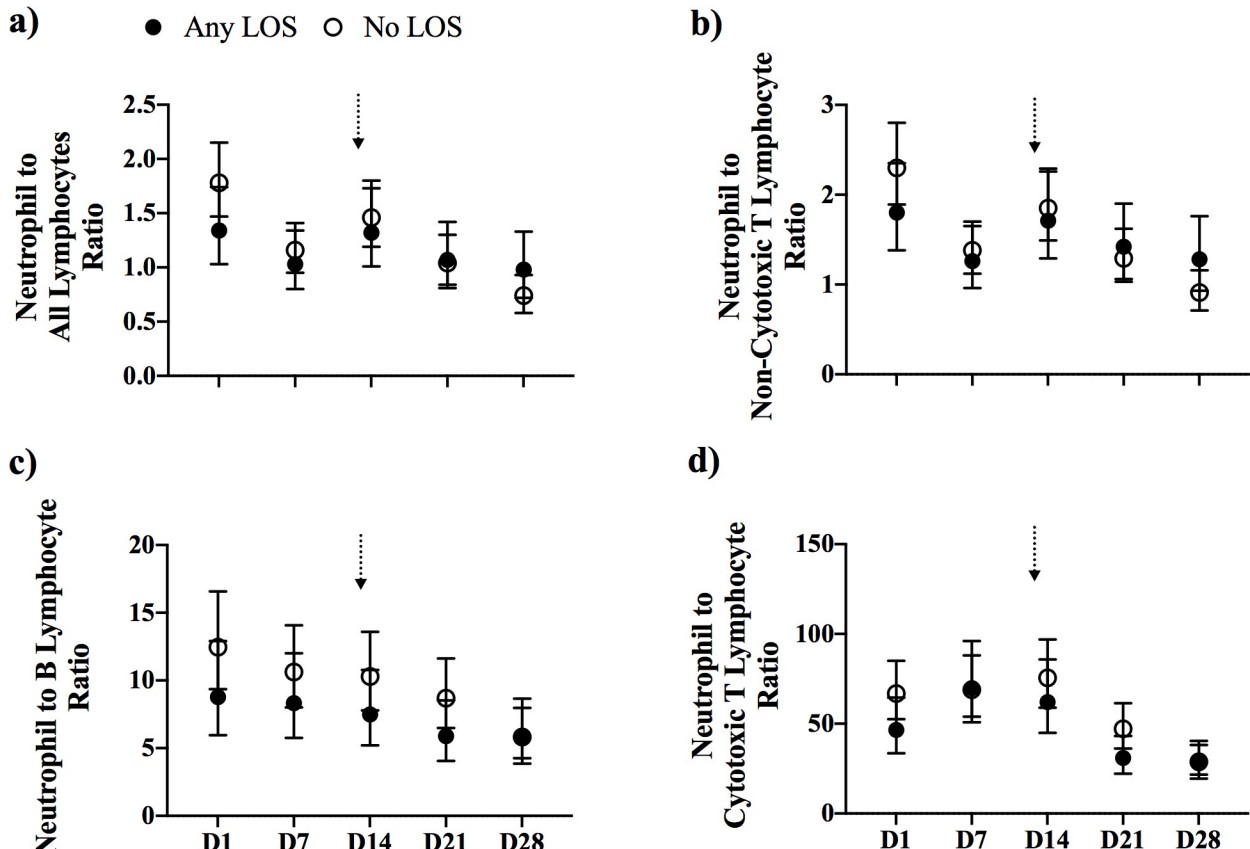

**Fig 3. Neutrophil-to-lymphocyte ratios.** The ratio of neutrophils to a) all lymphocytes, b) non-cytotoxic T lymphocytes, c) B lymphocytes, and d) cytotoxic T/NK lymphocytes, from preterm infants with any LOS (confirmed and clinical; closed circle) and no LOS (open circle). Analysis was performed by mixed linear regression analysis using a random effects model that was adjusted for known confounders, GA, birth weight z-score, sex and total leukocyte count. Data shown as geometric mean with 95% confidence intervals. Arrows indicate the median age of LOS onset of 13 days for infants in the any LOS group.

NK cells and B lymphocytes increased with age, similar findings to previous longitudinal studies that have characterised the T cell compartment in preterm infants [28, 32–34].

## Leukocyte populations in infants with and without LOS

We report that infants who later developed LOS had lower neutrophil counts on day 1 of age, albeit within the normal age-adjusted range [29]. Neutropenia, compounded by functional impairments, has been linked to sub-optimal microbial colonisation and LOS susceptibility in murine models of sepsis [35–38]. However, in human infant studies a link between early-onset neutropenia and LOS has not been confirmed [39–41]. Our observation of lower neutrophil counts in infants who subsequently develop LOS is transient and potentially affected by maternal factors (e.g. pre-eclampsia, placental insufficiency) that are associated with early-onset neutropenia in newborns [41].

Basophil counts and frequencies were elevated until day 14 and declined thereafter in infants without LOS, whereas they remained consistently lower in those who developed LOS. The role of basophils in human LOS remains unclear, with most data collected from animal models. Basophil-deficient mice exhibit reduced bacterial clearance and increased mortality in sepsis models, indicating an important role for basophils in initiating an immune response to

bacterial infection [42]. Our data of lower basophil counts in infants who developed LOS support a contribution of these cells to LOS susceptibility.

The elevated ratio of non-cytotoxic T to cytotoxic T/NK lymphocytes observed in our study is in keeping with the Th2-skewed response reported in early infancy [32, 33]. This skewing increases tolerance to pathogens, which is important for early life commensal colonisation, but may contribute to susceptibility to infection in infants [43]. Interestingly, we found that following LOS on days 21 and 28, preterm infants had reduced counts and frequencies of non-cytotoxic T lymphocytes compared to infants who did not develop LOS. Depleted circulating lymphocyte populations is a characteristic of sepsis-induced immunosuppression in adults, and is reported at post-mortem in preterm and term infants and children following fatal sepsis [44–46]. Taken together, these data suggest a degree of immune suppression following LOS in our cohort of preterm infants.

Compared to infants without LOS, immature granulocytes remained elevated near the time of LOS, a finding consistent with previous neonatal sepsis studies [23].

## Neutrophil-to-lymphocyte ratio

An elevated ratio of neutrophils to lymphocytes is a proposed biomarker for systemic inflammation in neonatal sepsis with an area under the curve of 0.78, sensitivity of 0.73 and specificity of 0.79 from receiver operating characteristic analysis [47]. We found that the ratio of neutrophils to lymphocytes was similar between preterm infants with and without LOS, indicating this ratio is a poor predictor of LOS in our cohort. The discrepancy in the ratio of neutrophils to lymphocytes in septic and non-septic infants between our study and those published, suggests that elevated ratios may not necessarily be present in the days or weeks leading up to and after LOS, likely due to the rapid inflammatory changes in response to infection [9]. Additionally, our leukocyte counts were adjusted known risk factors for LOS, which may refine the difference in neutrophil-to-lymphocyte ratio between infants with and without LOS.

## Study strengths and limitations

The strengths of this study include the longitudinal prospective sample collection through the first month of life and detailed analysis of leukocyte populations from peripheral blood in real time, both of which are unusual in preterm infant studies. Our analyses were adjusted for known risk factors for LOS [2]. We acknowledge that there are some limitations of the study, including the generalised classification of leukocytes and lack of specific populations, particularly in the T cell compartment (e.g. CD4, CD8, regulatory T cells and other innate-like T cells), that are known to play specific roles in the immune response to bacterial infection [31, 33, 34, 48, 49]. The sample size is modest, reflecting the difficulties of repeated standardised sampling in this population, which limits relating our laboratory findings to clinical outcomes and the influence of maternal factors and antibiotic therapy that may influence leukocyte populations. Moreover, the investigation into potential differences between Gram-positive and Gram-negative infections was limited by the modest number of infants with LOS particularly due to organisms other than Coagulase-negative staphylococci. Sample collection at the time of sepsis evaluation was not permitted due to ethical reasons, limiting analyses to weekly time points. The lack of functional data precludes a mechanistic interpretation of the relationship of leukocyte counts and frequencies to LOS. The correlation between the 5-fluorophore-6-marker flow cytometry method and manual/automated counts for some cell populations (e.g. basophils, monocytes) in blood samples from newborn infants is low and interpreting the counts in this study as a reference range should be done with caution [15, 16].

## Conclusions

This study adds to the limited knowledge of the composition of leukocyte populations in pre-term infant, particularly for those who develop LOS. Decreased non-cytotoxic T lymphocytes following LOS may reflect immunosuppression. Future studies further characterising leuko-cyte populations, particularly immunosuppressive populations (e.g. regulatory T lymphocytes and myeloid-derived suppressor cells), using additional markers and advanced techniques, such as high dimensionality analysis, that allow for in-depth detection beyond conventional gating, will provide complementary information on early life immune development in infants.

## Supporting information

**S1 File.**
(PDF)

## Acknowledgments

The authors would like to thank all participating families. We would also like to thank Gail Abernethy, Annie Chang, Chooi Heen Kok, and the clinical staff at King Edward Memorial Hospital Neonatal Directorate for assistance with recruitment and sample collection. We acknowledge the generosity of the Pathwest Clinical Immunology Laboratory, including Roslyn Hackshaw, Tracie Easter and Monica Kemp for provision of flow cytometry expertise and equipment. Lastly, laboratory assistance from Karen Riley, Stephanie Trend and Melissa Whinnen is acknowledged.

## Author Contributions

**Conceptualization:** Julie Hibbert, Tobias Strunk, Karen Simmer, Peter Richmond, David Burgner, Andrew Currie.

**Data curation:** Julie Hibbert.

**Formal analysis:** Julie Hibbert, Elizabeth Nathan, Dorota Doherty.

**Funding acquisition:** Tobias Strunk, Dorota Doherty, Karen Simmer, Peter Richmond, David Burgner, Andrew Currie.

**Investigation:** Julie Hibbert, Tobias Strunk, Amy Prosser, David Burgner, Andrew Currie.

**Methodology:** Julie Hibbert, Tobias Strunk, Amy Prosser, David Burgner, Andrew Currie.

**Project administration:** Julie Hibbert, Amy Prosser.

**Resources:** Karen Simmer, Peter Richmond, David Burgner, Andrew Currie.

**Supervision:** Tobias Strunk, Andrew Currie.

**Visualization:** Julie Hibbert.

**Writing – original draft:** Julie Hibbert.

**Writing – review & editing:** Julie Hibbert, Tobias Strunk, Elizabeth Nathan, Amy Prosser, Dorota Doherty, Karen Simmer, Peter Richmond, David Burgner, Andrew Currie.

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
