## [Decision Letter · Decision Letter 0]

3 Jan 2022

PONE-D-21-37106Composition of early life leukocyte populations in preterm infants with and without late-onset sepsisPLOS ONE

Dear Dr. Currie,

Thank you for submitting your manuscript to PLOS ONE. After careful consideration, we feel that it has merit but does not fully meet PLOS ONE’s publication criteria as it currently stands. Therefore, we invite you to submit a revised version of the manuscript that addresses the points raised during the review process.

We look forward to receiving your revised manuscript.

Kind regards,

Markus Sperandio

Academic Editor

PLOS ONE

Journal Requirements:

(This study was funded by grants from the National Health and Medical Research Council of Australia (#572548, www.nhmrc.gov.au), the Western Australia Telethon Channel 7 Trust and the Western Australia Department of Health.

DB is supported by a National Health and Medical Research Council of Australia Investigator Grant (#1175744, www.nhmrc.gov.au) and their research at the Murdoch Children's Research Institute is supported by the Victorian Government's Operational Infrastructure Support Program. 

TS is supported by a Raine Foundation and Western Australia Department of Health Clinician Research Fellowship (rainefoundation.org.au). 

JH is supported by a University Postgraduate Award and a Wesfarmers Centre of Vaccines and Infectious Diseases Scholarship (www.infectiousdiseases.telethonkids.org.au)

The funders had no role in the study design, data collection and analysis, decision to publish, or preparation of the manuscript.

(This study was supported by the National Health and Medical Research Council of Australia project grant (#572548), the Western Australia Department of Health, and the Western Australia Telethon Channel 7 Trust. TS is supported by a Raine Foundation and Western Australia Department of Health Clinician Research Fellowship. DB is supported by a National Health and Medical Council of Australia Investigator Grant (GTN1175744). Research at the Murdoch Children’s Research Institute is supported by the Victorian Government’s Operational Infrastructure Support Program. JH is supported by a University Postgraduate Award and postgraduate scholarship from the Wesfarmers Centre of Vaccines and Infectious Diseases. The funding sponsors had no role in the study design, collection, interpretation or analysis of data, writing the report or the decision to submit the article for publication.)

(This study was funded by grants from the National Health and Medical Research Council of Australia (#572548, www.nhmrc.gov.au), the Western Australia Telethon Channel 7 Trust and the Western Australia Department of Health.

DB is supported by a National Health and Medical Research Council of Australia Investigator Grant (#1175744, www.nhmrc.gov.au) and their research at the Murdoch Children's Research Institute is supported by the Victorian Government's Operational Infrastructure Support Program. 

TS is supported by a Raine Foundation and Western Australia Department of Health Clinician Research Fellowship (rainefoundation.org.au). 

JH is supported by a University Postgraduate Award and a Wesfarmers Centre of Vaccines and Infectious Diseases Scholarship (www.infectiousdiseases.telethonkids.org.au).

The funders had no role in the study design, data collection and analysis, decision to publish, or preparation of the manuscript.)

Please carefully address all concerns raised by the reviewers.

Reviewers' comments:

Reviewer's Responses to Questions

**Comments to the Author**

1. Is the manuscript technically sound, and do the data support the conclusions?

Reviewer #1: Yes

Reviewer #2: Yes

2. Has the statistical analysis been performed appropriately and rigorously? 

Reviewer #1: Yes

Reviewer #2: Yes

3. Have the authors made all data underlying the findings in their manuscript fully available?

Reviewer #1: Yes

Reviewer #2: Yes

4. Is the manuscript presented in an intelligible fashion and written in standard English?

Reviewer #1: Yes

Reviewer #2: Yes

5. Review Comments to the Author

Reviewer #1: This is a clearly designed study to characterize the circulating leukocyte subsets in preterm neonates. The authors performed weekly FACS analyses during the first month of life and found differences between those infants that developed late onset sepsis compared to those who did not develop late onset sepsis.

Major criticism

• The staining panel is relatively simplified. Why did the authors not look at B1 B cells or other subsets that are highly interesting in the newborn? The atypical selection of leukocyte populations should be explained.

• Why was LOS defined retrospectively? It would have been much better to define this a priori.

• Definition of clinical LOS is missing: Which clinical signs were documented?

Minor comments

• The authors mention that they collected 0,8 ml blood samples but they used only 50µl for staining. How does this come?

• How many infants would have been eligible and how many infants were recruited. Were ther potential sources of a biased recruitment?

• The authors should discuss whether the administration of antibiotics may have an influence on leukocyte subpopulations.

• I do not understand the sense of the arrows in Figure 1.

Reviewer #2: The manuscript "Composition of early life leukocyte populations in preterm infants with and without late-onset sepsis" by Hibbert et al. present a descriptive single centre cohort study of 119 preterm infants below 30 weeks of gestation. The study analysed leukocyte composition of infants during the first month of life. Although data exist on this topic, the manuscript impresses by the rather large number of infants included and the systematic longitudinal follow up.

Thus, the analysis of this cohort is of interest, and with the thoughts below, the authors should be encouraged to answer some more of possibly relevant questions:

1) All Figures: For better understanding: please explain in more detail the group allocation at the time points given. Is “No LOS” at a certain time point all subjects having had no LOS until this time point or during the whole study period or only until this time point? And includes “LOS” at a time points all subjects having had LOS before or up to this time point or during the whole study period? If group allocation has been made on basis of the whole study period, the authors should analyse the data allocating the subjects to “No Los” and “LOS” before and after sepsis.

2) For all Figures: please explain “adjustment” in more detail.

3) Page 9, line 164: what about other confounders: steroids, mode of delivery, antibiotics?

4) Table 1 and page 10, line 185: age of LOS is given as 13 (10-18) in the table, and 13 (6-26) in the text, please correct if necessary.

5) What about the influence of pathogens? The authors should be encouraged to analyse the difference of Gram-positive and Gram-negative sepsis and especially of CONS-sepsis.

6. PLOS authors have the option to publish the peer review history of their article (what does this mean?). If published, this will include your full peer review and any attached files.

Reviewer #1: No

Reviewer #2: **Yes: **Christian Gille, MD

---

## [Author Response · Author response to Decision Letter 0]

1 Feb 2022

Reviewer 1

This is a clearly designed study to characterize the circulating leukocyte subsets in preterm neonates. The authors performed weekly FACS analyses during the first month of life and found differences between those infants that developed late onset sepsis compared to those who did not develop late onset sepsis.  R1. Major criticism The staining panel is relatively simplified. Why did the authors not look at B1 B cells or other subsets that are highly interesting in the newborn? The atypical selection of leukocyte populations should be explained.

Response: We acknowledge that the antibody panel used in this study is dated however this study was conducted between 2009 and 2011, a time when this panel was relatively novel. We agree that it would be ideal to have used a more sophisticated panel that included additional leukocyte sub-populations. However, a limited number of fixable fluorochromes and available channels (6-channels) on our flow cytometer were available at the time of the study. This 6-marker/5-colour surface marker panel for an extended white cell differential was therefore the best option at the time, especially in acutely ill individuals. Beckman Coulter since adapted this panel as their CytoDiff reagent, which has been validated in various health conditions (PMID: 21875393; 21779183), including infant cord blood and peripheral blood from critically ill newborns (PMID: 22862853; 27834108). The limitations of this panel have been raised in the discussion on page 28. We agree that a range of more recently identified innate lymphocytes and adaptive regulatory cells would be of great interest for future studies in the field, potentially with higher-throughput sequencing-based technologies. However, we still believe that our study adds important knowledge of postnatal extended leukocyte subset development in very preterm infants.

R1.1. Why was LOS defined retrospectively? It would have been much better to define this a priori.

Response: For this observational study, we followed our local NICU guidelines (blood culture positivity and elevated, serially measured C-reactive protein) for identifying late-onset sepsis. Therefore, late-onset sepsis was defined at the time of sepsis based on pre-defined criteria and not retrospectively. 

We have removed ‘retrospectively’ (line 93 of the original submission) from the methods section to avoid confusion.

 R1.2. Definition of clinical LOS is missing: Which clinical signs were documented?

Response: Clinical LOS was defined as a negative blood culture and a CRP of > 15mg/L within 72 hours of blood culture (line 96 of the original submission). Blood culture was performed if infants had clinical signs suggestive of LOS, including increased ventilation or oxygen requirements, lethargy, decreased perfusion, temperature instability, vomiting, feed intolerance, increased apnoea and/or bradycardia. 

The Methods section has been updated to include this information.

Minor comments R1.3. The authors mention that they collected 0,8 ml blood samples but they used only 50µl for staining. How does this come?

R1.3 Response: Thank you for raising this point. The samples in the manuscript were part of a prospective, longitudinal observational cohort study of innate immune system development in very preterm infants. The 0.8 mL of blood collected in this study was used in several assays for characterising immune development, including the assays reported in PMID: 31960030, 31578034, and 29746597. In addition to these published assays, several assays have not been published yet, including additional FACS panels for monocytes/DCs and T lymphocytes, as well as several whole blood pathogen phagocytosis and killing assays. Collectively, these analyses use a total of 0.8 mL of blood. We are currently analysing these data for manuscript preparation in 2022.

R1.4. How many infants would have been eligible and how many infants were recruited. Were there potential sources of a biased recruitment?

R1.4 Response: All infants born < 30 weeks gestational age who were admitted to the NICU were eligible for recruitment unless major congenital anomalies were known. Of the 398 eligible infants during the study period, 129 were recruited. The basic demographic and clinical data were similar between the eligible infants who were and were not recruited to the study (GA 26.9 vs 26.83 weeks, p=0.76; birth weight 942 vs 962 g, p=0.54; males 54 vs 52%, p=0.62; caesarean section 57% vs 65%, p=0.914). Therefore, there is no indication of biased recruitment.

Please see table in PDF appended to the end of the PDF.

R1.5. The authors should discuss whether the administration of antibiotics may have an influence on leukocyte subpopulations.

R1.5 Response: We agree that this is a topic of interest since some evidence from neonatal animal models of infection shows exposure to antibiotics, particularly exposure in utero, can alter the function, phenotype, and frequency of leukocytes (PMID 27036912, 22705104; 33178650). However, our study was not designed or powered to assess the potential influence of antibiotic use on leukocyte counts and therefore is outside the scope of this manuscript. 

The ‘strengths and limitations’ section has been updated.

R1.6. I do not understand the sense of the arrows in Figure 1.

R1.6 Response: Thank you for raising that the illustration is unclear. The arrows in Figure 1 represent the median age (i.e., 13 days) that infants in the ‘any LOS’ group developed any LOS. 

The figure legend has been updated to make this clearer.

Reviewer 2

The manuscript "Composition of early life leukocyte populations in preterm infants with and without late-onset sepsis" by Hibbert et al. present a descriptive single centre cohort study of 119 preterm infants below 30 weeks of gestation. The study analysed leukocyte composition of infants during the first month of life. Although data exist on this topic, the manuscript impresses by the rather large number of infants included and the systematic longitudinal follow up. Thus, the analysis of this cohort is of interest, and with the thoughts below, the authors should be encouraged to answer some more of possibly relevant questions:

 R2.1. All Figures: For better understanding: please explain in more detail the group allocation at the time points given. Is “No LOS” at a certain time point all subjects having had no LOS until this time point or during the whole study period or only until this time point? And includes “LOS” at a time points all subjects having had LOS before or up to this time point or during the whole study period? If group allocation has been made on basis of the whole study period, the authors should analyse the data allocating the subjects to “No Los” and “LOS” before and after sepsis.

R2.1 Response: Thank you for your input and consideration. Group allocations were made based on the whole study period, that is, infants that developed LOS during the study period were assigned to the LOS group from Day 1 so that the trajectory of their immune development before and after the LOS episode could be tracked. Peripheral blood was collected for this study was on days 1, 7, 14, 21 and 28 of life. Due to ethical and clinical care reasons, we could not collect an additional sample at the time of sepsis onset and therefore were unable to perform leukocyte counts at the time of sepsis. The primary aims of this study were to i) characterise the innate immune responses in a cohort of very preterm infants during the period of greatest risk of LOS, and ii) evaluate if any immunological features preceding an episode of LOS may predict the risk of LOS occurring. Therefore, we elected to allocate infants to their respective groups for the whole study period. As expected from our own and published data, 61% of LOS cases occurred between days 7 and 14, a further 31% between days 15 and 21, and only two infants each between days 1-7 and 22-28, respectively. Therefore, the differences observed between infants with and without LOS after day 7 are valid for 92% of infants. However, we agree that data leading up to, during, and after sepsis are important, and we feel future studies should consider this approach which will further advance our knowledge in this area.

The Statistical Analysis section on page 9 has been updated with this information.

R2.2. For all Figures: please explain “adjustment” in more detail.

R2.2 Response: Our statistical models (mixed linear regression analysis using random effects) were adjusted for known confounders, GA, birth weight z-score, sex and total leukocyte count (page 9 under the Statistical Analysis section). 

The figure legends on pages 13 and 14 have been updated to include this information.

 R2.3. Page 9, line 164: what about other confounders: steroids, mode of delivery, antibiotics?

R2.3 Response: We thank the reviewers for this suggestion however we do not think it is necessary to include these confounders in our model for the following reasons: 

i) Over 90% of mothers of infants in the any LOS and no LOS groups received antenatal steroids (Table 1; 93% and 97%, p=0.35), therefore we would not expect this to influence the results. Moreover, antenatal steroids are not associated with an increased risk of LOS when gestational age is corrected for and with the proven benefit of antenatal steroids, it would be unethical to withhold treatment from women at risk of preterm birth (PMID: 33368142, 26142898, 1171763730646235). Lastly, it would not be feasible to recruit a sufficiently large number of very preterm infants not exposed to antenatal steroids.

ii) The percent of caesarean sections was similar between the any LOS and no LOS groups (caesarean section 58% and 55%, p=0.85), as shown in Table 1; therefore, we would not expect the mode of delivery to affect our results.

iii) Postnatal steroid use was higher in our cohort of infants with any LOS compared to the no LOS infants (18% and 4%, p=0.005). However, postnatal steroids were given after the study period and therefore would not affect our results. Consistent with other published neonatal studies a higher percentage of infants in the any LOS who received postnatal steroids is expected because infants who develop LOS are typically younger, smaller and more unwell than uninfected infants (PMID: 12165580). Moreover, LOS itself drives inflammation and increases the risk of chronic lung disease, which is treated using postnatal steroids (PMID: 31664055, 10585987). The ‘strengths and limitations’ section has been updated.

 R2.4. Table 1 and page 10, line 185: age of LOS is given as 13 (10-18) in the table, and 13 (6-26) in the text, please correct if necessary.

R2.4 Response: Thank you for your careful review. At first glance, these numbers look contradictory; however, the text reports the range and the table reports the interquartile range (25th – 75th). 

The text under the Results section on page 10 has been updated to include the interquartile range to avoid confusion.

---

## [Decision Letter · Decision Letter 1]

17 Feb 2022

Composition of early life leukocyte populations in preterm infants with and without late-onset sepsis

PONE-D-21-37106R1

Dear Dr. Currie,

We’re pleased to inform you that your manuscript has been judged scientifically suitable for publication and will be formally accepted for publication once it meets all outstanding technical requirements.

Kind regards,

Markus Sperandio

Academic Editor

PLOS ONE

Additional Editor Comments (optional):

The authors have sufficiently addressed the comments raised by both reviewers.

Reviewers' comments:

Reviewer's Responses to Questions

**Comments to the Author**

1. If the authors have adequately addressed your comments raised in a previous round of review and you feel that this manuscript is now acceptable for publication, you may indicate that here to bypass the “Comments to the Author” section, enter your conflict of interest statement in the “Confidential to Editor” section, and submit your "Accept" recommendation.

Reviewer #1: All comments have been addressed

2. Is the manuscript technically sound, and do the data support the conclusions?

Reviewer #1: Yes

3. Has the statistical analysis been performed appropriately and rigorously? 

Reviewer #1: Yes

4. Have the authors made all data underlying the findings in their manuscript fully available?

Reviewer #1: Yes

5. Is the manuscript presented in an intelligible fashion and written in standard English?

Reviewer #1: Yes

6. Review Comments to the Author

Reviewer #1: The authors have addressed all reviewer's comments. Although the study has some weaknesses, it still adds new insights that are worth to be reported.

7. PLOS authors have the option to publish the peer review history of their article (what does this mean?). If published, this will include your full peer review and any attached files.

Reviewer #1: No

---

## [Editor Report · Acceptance letter]

21 Feb 2022

PONE-D-21-37106R1 

Composition of early life leukocyte populations in preterm infants with and without late-onset sepsis 

Dear Dr. Currie:

I'm pleased to inform you that your manuscript has been deemed suitable for publication in PLOS ONE. Congratulations! Your manuscript is now with our production department. 

Kind regards, 

on behalf of

Prof. Dr. Markus Sperandio 

Academic Editor

PLOS ONE